# Pain Processing in Older Adults and Its Association with Prefrontal Characteristics

**DOI:** 10.3390/brainsci10080477

**Published:** 2020-07-24

**Authors:** Steffie Bunk, Mónica Emch, Kathrin Koch, Stefan Lautenbacher, Sytse Zuidema, Miriam Kunz

**Affiliations:** 1Department of General Practice and Elderly Care Medicine, University Medical Center Groningen, University of Groningen, 9713GZ Groningen, The Netherlands; s.u.zuidema@umcg.nl (S.Z.); miriam.kunz@med.uni-augsburg.de (M.K.); 2Department of Neuroradiology, Klinikum rechts der Isar, Technische Universität München, 81675 Munich, Germany; monica.emch@gmail.com (M.E.); kathrin.koch@tum.de (K.K.); 3Graduate School of Systemic Neurosciences, Ludwig-Maximilians-Universität München, 82152 Martinsried, Germany; 4Physiological Psychology, University of Bamberg, 96045 Bamberg, Germany; stefan.lautenbacher@uni-bamberg.de; 5Department of Medical Psychology and Sociology, University of Augsburg, 86159 Augsburg, Germany

**Keywords:** pain inhibition, executive functioning, magnetic resonance imaging, voxel-based morphometry, tractography

## Abstract

Aging is known to affect nociceptive processing, e.g., the ability to inhibit pain. This study aims to investigate whether pain responses in older individuals are associated with prefrontal characteristics, namely (i) executive functioning performance and (ii) structural brain variations in the prefrontal cortex. Heat and pressure stimuli were applied to assess pressure pain sensitivity and endogenous pain inhibition in 46 healthy older individuals. Executive functioning performance was assessed in three domains (i.e., cognitive inhibition, shifting, and updating) and structural brain variations were assessed in both gray and white matter. Overall pain responses were significantly associated with the executive functioning domains cognitive inhibition and shifting. However, no specific type of pain response showed an especially strong association. Endogenous pain inhibition specifically showed a significant association with gray matter volume in the prefrontal cortex and with variations in white matter structure of tracts connecting the prefrontal cortex with the periaqueductal gray. Hierarchical regression analyses showed that these variations in the prefrontal cortex can explain variance in pain inhibition beyond what can be explained by executive functioning. This might indicate that known deficits in pain inhibition in older individuals are associated with structural variations in prefrontal areas.

## 1. Introduction

Acute and chronic pain are more common in older than in younger individuals [1,2], and it has been suggested that this increased pain prevalence is partly due to age-related changes in nociceptive processing. Using experimental pain to study age-related changes in pain processing, the majority of evidence point to an increase in pain threshold (indicating decreased nociceptive processing), but also an increase in temporal summation of pain as well as a decrease in endogenous pain inhibition (both indicating increased nociceptive processing). Based on these findings, it has been suggested that both ascending excitatory and descending inhibitory pathways are altered during aging. However, inhibitory pathways might be more affected than excitatory pathways, thus leading to a higher pain prevalence in older individuals [3].

Previous studies have tried to look for mechanisms that might underlie or accompany these age-related changes in pain processing. One of the mechanisms that has been studied is cognitive functioning, in particular, executive functioning. Executive functions are higher order skills that are linked to the function of the frontal cortex and that enable an individual to control behavior [4]. Frontal functions are often subdivided into three domains: cognitive inhibition, shifting, and updating [5]. Poorer functioning in these domains was found to be associated with reduced pain tolerance [6], increased pain responses [7], and poorer endogenous pain inhibition [8,9] in older individuals. Thus, frontal functioning might be one mechanism underlying increased pain processing in older individuals.

Executive functioning tests are considered to be an indirect measure of frontal functioning. Another indirect measure of frontal functioning is to assess the structure of the frontal cortex. It is known that the structure of the frontal cortex changes during aging, as observed by gray matter atrophy [10] and alterations in white matter structure [11]. Given that the prefrontal cortex is thought to play a key role in the processing of pain, especially in descending pain modulation [12], we hypothesize that pain responses in older individuals are not only associated with executive functions but also with structural variations within (gray matter) and between (white matter) the prefrontal cortex and other brain regions involved in pain processing. 

To test this hypothesis, we assessed pain responses in a group of older individuals using two experimental pain paradigms, namely, sensitivity to phasic pressure pain and conditioned pain modulation (CPM) using heat and pressure pain to assess endogenous pain inhibition. Responses were assessed via verbal ratings as well as via facial expression to capture two output channels of pain. Assessing two output channels (self-report as the more controlled output where older adults have been found to be more reticent in reporting pain and facial expression as the more automated output with no age-related changes [13,14]) might allow a more comprehensive assessment of pain. We investigated whether these pain responses in older individuals are associated with prefrontal characteristics by relating them (i) to the different domains of executive functioning and (ii) to variations in gray and white matter structure of the prefrontal cortex.

## 2. Materials and Methods

### 2.1. Participants

Fifty-nine healthy older individuals were recruited through advertisements and among students of the local University of the Third Age. After exclusion of 13 participants of whom we could not obtain a structural brain scan (because of claustrophobia (*N* = 2), metal implants (*N* = 9), or image artifacts (*N* = 2)), forty-six individuals participated in this study. None of the participants had a history of major neurological or psychological disorders and no participants had taken analgesic medication on the day of testing. The study was conducted in accordance with the Declaration of Helsinki and approved by the ethics committee of the University Medical Center Groningen (code 2016/398). A written informed consent was obtained from all participants. Participants received monetary compensation for their participation. 

### 2.2. Study Procedure

The study consisted of three parts. Part one (experimental pain tests) and part two (executive functioning tests) were generally conducted between 10:00 a.m. and 3:00 p.m. On a separate day, participants underwent a structural magnetic resonance imaging (MRI) scan (part three). Figure 1 provides an overview of all variables used in the three parts of the study.

### 2.3. Experimental Pain Protocol

#### 2.3.1. Apparatus

Pressure stimuli were applied to the midpoint of the upper border of the trapezius muscle (back shoulder area) using a pressure algometer with a probe area of 1 cm^2^ (Algometer type II, Somedic Sales AB, Hörby, Sweden). Heat stimuli were administered to the right inner forearm using a thermal sensory analyzer (Medoc TSA II, Ramat-Yishai, Israel) with a Peltier thermode with a stimulation surface of 6 cm^2^. 

#### 2.3.2. Pressure Pain Sensitivity

Four different pressure intensities (50, 200, 400, and 500 kPa) were applied to the shoulder in an ascending order. There were two trials, one to the right shoulder and one to the left shoulder. An ascending order was chosen to reduce anxiety in participants as well as to be able to immediately stop with the stimulation protocol if a participant found the stimulation to be too painful (this did not occur in the present sample). Pressure was increased steadily for 2 s until the desired intensity was reached and was then kept constant for 5 s.

#### 2.3.3. CPM 

Endogenous pain inhibition was assessed using the CPM paradigm, in which a conditioning stimulus and a test stimulus are applied simultaneously to different parts of the body to test the modulating effect of the conditioning stimulus. The test stimuli used in this CPM paradigm were comparable to the stimuli used to assess pressure pain sensitivity, namely, four pressure intensities of 50, 200, 400, and 500 kPa applied to the right and left shoulder. Tonic heat stimulation served as the conditioning stimulus and consisted of a series of small heat pulses at a frequency of 30 pulses/min with an amplitude of 1.3 °C applied on the right inner forearm [15,16,17,18,19]. Thus, the test stimuli were presented bilateral and the conditioning stimulus was presented on the right side. In the first block, the test stimuli on the shoulder were applied together with nonpainful tonic heat stimulation of 43 °C (baseline stimulus). In the second block, the test stimuli on the shoulder were applied together with painful tonic heat stimulation of 45 °C (conditioning stimulus). The blocks lasted for around 2 min. Pain inhibition was indicated by a lower pain response to pressure stimuli paired with painful heat than to pressure stimuli paired with nonpainful heat (pressure pain during painful heat − pressure pain during nonpainful heat ≤ 0). Pain facilitation was indicated by a higher pain response to pressure stimuli paired with painful heat than to pressure stimuli paired with nonpainful heat (pressure pain during painful heat − pressure pain during nonpainful heat ≥ 0). 

#### 2.3.4. Pain Responses

##### Ratings

Immediately after each stimulus (pressure and heat stimulation), participants were asked to rate the pain sensation using a five-category verbal rating scale (no pain, mild pain, moderate pain, strong pain, and very strong pain). For further analyses, the ratings of the pressure stimuli on the right and left shoulder were averaged to obtain one rating per intensity. 

##### Facial responses

Facial responses were assessed during each pressure stimulus. The heat stimulation serving as the conditioning stimulus in the CPM protocol was only mildly painful and therefore did not elicit facial responses. A camera was placed approximately 2 m in front of the participants to videotape the faces of the participants. Participants were instructed to look into the camera and were asked not to talk when pain was induced. Facial responses were analyzed using the Facial Action Coding System (FACS) [20]. This system describes 44 visually distinguishable action units (AUs). A FACS coder (qualified by passing the FACS examination) both identified the frequency and intensity of all AUs that occurred during stimulation. Another qualified FACS coder recoded a subset of 10% of the video segments. Inter-rater reliability, calculated using the Ekman–Friesen formula (number of AUs agreed upon × 2 and divided by the overall amount of AUs coded), was 0.80, which compares favorably with other research in the FACS literature [21,22]. The intensity of each AU was scored using a 5-point scale, which was entered into a time-related database from the onset of the stimuli till the end of the stimuli (5 s) using the Observer Video-Pro (Noldus Information Technology, Wageningen, The Netherlands). Some AUs represent facial movements of the same muscle and were, therefore, combined to reduce the number of variables (AU 1/2, AU 6/7, AU 9/10, and AU 25/26/27). For further analyses, we only used the AUs that occurred during at least 5% of the painful stimuli and that occurred more frequently during painful pressure stimulation (500 kPa) than during nonpainful pressure stimulation (50 kPa) (Cohen’s d effect size d ≥ 0.5) [23,24]. These were AU 1/2, AU 4, AU 6/7, AU 9/10, and AU 25/26/27 (listed in Table 1). To obtain a FACS composite score for each stimulus intensity, the frequency of each AU was multiplied by the AU intensity score and then averaged over all AUs. 

### 2.4. Executive Functioning Tests 

Three paper-and-pencil neuropsychological tests were used to assess executive functioning performance in the domains cognitive inhibition, updating, and shifting [5]. 

#### 2.4.1. Cognitive Inhibition 

The Stroop Interference score, measured using the Stroop Color and Word test [25], was used to assess cognitive inhibition. This tests consists of three parts: (i) the Reading task, in which participants are asked to read aloud color names, (ii) the Color Naming task, in which participants are asked to name the color of printed colors, and (iii) the Color Word task, in which participants are asked to name the printed color of a color word that is printed in a different color than the meaning of the word. Thus, in this last task the participants must inhibit naming the written word. Every part of the test consisted of 100 items and lasted for 45 s. The Stroop Interference score was defined by the difference in number of correct responses between the Color Naming task and the Color Word task. A higher interference score indicated worse cognitive inhibition. 

#### 2.4.2. Shifting 

Shifting was assessed using the Trail Making Test Part B, which measures the ability to alternate between cognitive categories [26]. Participants were asked to connect randomly placed numbers (1–13) and letters (A–L) in ascending order, alternating between numbers and letters. The score of the Trail Making Test Part B was the time to complete the task. 

#### 2.4.3. Updating 

Updating was assessed using the Letter Fluency task [27], in which participants are asked to produce as many words in Dutch starting with the letter D within 1 min. Thus, this task measures the ability to retrieve words and keep track of those words to avoid repetition. The score of this task was the total number of words produced. 

### 2.5. MRI Acquisition and Analysis

MR images were acquired using a 3T MRI scanner (Magnetom Prisma, Siemens Healthineers, Erlangen, Germany). A T1-weighted image was acquired using a magnetization prepared rapid acquisition gradient echo (MPRAGE) sequence (176 sagittal slices, repetition time = 2300 m/s, echo time = 2.98 m/s, inversion time = 900 m/s, flip angle = 9°, voxel size = 1 × 1 × 1 m^2^, and field-of-view = 256 m^2^). Diffusion-weighted images were acquired along 64 directions using a *b*-value of 1000 s/m^2^ (60 slices, repetition time = 6300 m/s, echo time = 66 m/s, voxel size is 2.2 × 2.2 × 2.2 m^3^, and field-of-view = 220 m^2^). Ten volumes with no diffusion weighting (*b*-value of 0 s/m^2^) were acquired, one at the beginning and nine at the end of the acquisition. Despite of claustrophobia being an exclusion criterion, two participants experienced claustrophobia halfway during scanning time and, therefore, the acquisition of the diffusion-weighted data could not be completed.

#### 2.5.1. Voxel-Based Morphometric Analyses of Gray Matter

The T1-weighted images were used to identify significant correlations between pain responses and regional gray matter volume using voxel-based morphometry (VBM), a voxel-wise analysis of gray matter volume [28]. VBM analyses were performed using the Computational Anatomy Toolbox 12 (Jena University Hospital, Jena, Germany) within Statistical Parametric Mapping (SPM) 12 (Wellcome Department of Cognitive Neurology, London, UK), implemented in MATLAB (MathWorks, Natick, MA, USA). The VBM procedure involves spatial normalization of the MPRAGE images by registering the images to standard space (to correct for global differences in brain shape) and segmentation into gray matter images. After a data quality check, these gray matter images were smoothed using the default 8 mm full-width at half maximum Gaussian kernel.

First, exploratory whole brain VBM analyses were performed. Second, VBM analyses were restricted to areas in the frontal cortex that were associated with pain sensitivity and pain inhibition in previous structural and functional MRI studies [29,30,31,32,33,34] using the small volume correction function implemented in SPM. The following regions of interest were used: the ventromedial prefrontal cortex (vmPFC; defined as the combination of Brodmann areas (BA) 10 and 11), the dorsolateral prefrontal cortex (dlPFC; BA 9 and 46), and the ventrolateral prefrontal cortex (vlPFC; BA 44, 45, and 47). The masks for these regions of interest were defined using the Talairach Daemon atlas in the WFU PickAtlas toolbox [35,36].

#### 2.5.2. Probabilistic Tractography of White Matter

Probabilistic tractography can trace white matter tracts in the brain using diffusion-weighted MR data. This is possible because diffusion-weighted MRI allows to measure diffusivity of water molecules within axons [37]. Given that diffusivity of water is less restricted along an axon than perpendicular to an axon, it is possible to estimate fiber orientations. One measure to quantify this orientation is fractional anisotropy (FA). An FA value of 0 indicates unrestricted diffusion (isotropic diffusion), while an FA value of 1 indicates diffusion in only one direction (anisotropic diffusion).

In this study, we aimed to investigate whether experimental pain responses in older individuals correlate with FA of the white matter tracts between the prefrontal cortex and the periaqueductal gray (PAG), located in the brainstem. The PAG has direct anatomical connections with the prefrontal cortex [38] and sends neuronal input to key components of the descending pain modulatory system, including the rostroventromedial medulla, caudal ventrolateral medulla, and dorsal reticular nucleus [39]. Stein et al. (2012) have demonstrated that FA of the white matter tracts connecting the prefrontal cortex with the PAG correlates with placebo analgesia [40]. We followed their method, aiming to investigate whether pressure pain sensitivity and CPM in older individuals also correlate with FA of the white matter tracts connecting the prefrontal cortex with the PAG. Stein et al. only studied the tracts between the PAG and the dlPFC, but we extended our analysis to the white matter tracts between the PAG and the vlPFC as well as the vmPFC, to mirror our analysis approach for gray matter.

All diffusion-weighted imaging analyses were performed using the FMRIB Software Library (FSL version 6.0.0, Oxford, UK). Preprocessing of the images consisted of visual inspection, motion correction, correction of eddy current distortions, and removal of nonbrain tissue. The orientation of the fibers in each voxel was estimated using BEDPOSTX (Bayesian Estimation of Diffusion Parameters Obtained using Sampling Techniques with Crossing Fibers), which models the orientation of multiple crossing fibers within each voxel [41,42]. PROBTRACKX (probabilistic tracking with crossing fibres) was used to generate 5000 pathways from every voxel within the PAG, each following a slightly different route based on the fiber orientations. We used the default options of BEDPOSTX and PROBTRACKX. Because there is no PAG mask available, the mask of the PAG was manually drawn based on Montreal Neurological Institute (MNI) coordinates of previous studies investigating the PAG [43]. Only the tracts that passed the prefrontal radiation of the thalamus [40] and either the dlPFC, vlPFC (left and right), or vmPFC were retained for further analysis. The generated pathways were terminated as soon as the pathways left the prefrontal cortex. The mask of the thalamus was created using the Oxford Thalamic Connectivity Probability Mask within FSL. The dlPFC, vlPFC, and vmPFC masks were defined by the middle and inferior frontal gyrus of the Harvard-Oxford Cortical Structural Atlas [38]. Using the Talairach Daemon atlas, we verified that the premotor cortex (BA 6) was not included in the masks. All masks were made in standard space. Therefore, registration from diffusion space to standard space was performed within the FSL diffusion toolbox, which results in transformation matrices needed for PROBTRACKX. By default, output files of PROBTRACKX are produced in the same space as the original masks, in this case, standard space. The original diffusion FA images were linearly registered to standard space and up-sampled to a voxel size of 1 mm^3^ using trilinear interpolation to match the PROBTRACKX output files [44].

Probabilistic tractography results in a connection probability map in which the value of every voxel represents the number of generated pathways that pass through that voxel, i.e., the probability that a voxel is connected to the PAG and the prefrontal cortex. The results were normalized by dividing the number of generated pathways by the total number of generated pathways and then thresholded at 0.01% [45,46]. As a last step, the mean FA from voxels within the thresholded pathways between the PAG and each of the five prefrontal regions (dlPFC left, dlPFC right, vlPFC left, vlPFC right, and vmPFC) was extracted for each participant.

### 2.6. Statistical Analysis

All statistical analyses were performed using IBM SPSS Statistics 24, except for the VBM analyses, which were performed using SPM 12.

Preanalyses/pain responses in older individuals: As a first step, we used analyses of variance to examine whether pain responses were affected by potential confounders, namely, age, sex, and level of education, which have been shown in previous studies to impact pain [47]. The second analysis step aimed at describing the pain responses in the present sample. We, therefore, used analyses of variance to test whether subjective and facial responses significantly increased across pressure intensities. Paired-samples *t*-tests were used to test whether participants showed a CPM effect, defined by the difference between the response to the test stimulus combined with the conditioning stimulus and the test stimulus combined with the baseline stimulus.

Hypotheses testing:

Association between pain responses and executive functioning: Three multiple regression analyses were conducted to examine whether variance in pain responses could be explained by the executive functioning domains cognitive inhibition, shifting, and updating. The scores of the executive functioning tests were entered separately as predictor variables and the pain responses were entered as dependent variables. Univariate tests were used to examine which specific pain response could be predicted by the executive functioning tests.

Association between pain responses and gray matter volume: To examine whether pain responses are associated with regional gray matter volume in the brain, the smoothed normalized gray matter images resulting from VBM were entered in a general linear model. Pain responses were entered separately as covariates of interest. In addition, total intracranial volume was entered as a covariate in each general linear model to correct for differences in brain size. Exploratory whole brain analyses as well as region of interest analyses were performed. The threshold for significance was set at *p* < 0.05 family-wise error (FWE) corrected at peak level for both whole brain analyses as well as region of interest analyses.

Association between pain responses and white matter structure: To examine whether variance in pain responses could be explained by the structure of the white matter tracts between the PAG and the vlPFC, dlPFC (both left and right), and vmPFC, stepwise forward regression analyses were conducted. The mean FA values within these white matter tracts were entered as predictors and the pain responses were entered separately as dependent variables.

## 3. Results

### 3.1. Pain Responses in Older Individuals

Experimental pain responses were not significantly affected by age (self-report: F(1,44) = 0.28, *p* = 0.60; facial expression: F(33.01,54.37) = 1.62, *p* = 0.057) nor by level of education (self-report: F(4,40) = 0.54, *p* = 0.706; facial expression: F(1,44) = 0.52, *p* = 0.47). However, pain responses differed significantly between men and women, with women having increased responses to the pressure stimulation compared to men. For this reason, demographic characteristics, executive functioning performance and pain responses are displayed separately for men and women in Table 2.

Pressure pain sensitivity: As can be seen in Figure 2A, pain ratings increased with increasing pressure intensity (F(2.11,92.71) = 283.25, *p* < 0.001). We also found a significant increase in facial responses across the different intensities of pressure stimuli (F(2.07,91.11) = 13.90, *p* < 0.001), as can be seen in Figure 2B. The pressure stimuli of 400 and 500 kPa were on average rated as painful (verbal rating scale ≥ 1; Figure 2A). Due to the low variability in pain ratings to these stimuli (ratings varying mostly between mild to moderate pain; see also Table 2), we decided to define pressure pain sensitivity as the average response of the 400 and 500 kPa stimuli. Given the found sex differences in Table 2, we also tested for sex difference for these combined scores and found significant differences (self-report: *t* (44) = −2.61, *p* = 0.012; facial expression: *t* (25.54) = −3.55, *p* = 0.002). Sex was, therefore, added as a covariate in further analyses on pressure pain sensitivity.

CPM: The pressure stimuli paired with painful heat were not rated significantly less painful (Figure 2A) or elicited significantly weaker facial responses (Figure 2B) than the pressure stimuli paired with nonpainful heat. Thus, no significant CPM effects were observed. When looking at the descriptive data, less than half of the participants showed pain inhibition (15.2% for self-report and 45.7% for facial expression). Again, we tested for potential sex differences. CPM was not significantly different between men and women (self-report: *t* (44) = 1.66, *p* = 0.10; facial expression: *t* (27.94) = 1.11, *p* = 0.28), and we, therefore, refrained from entering sex as covariate in further CPM-related analyses.

### 3.2. Predictors of Pain Responses in Older Individuals

#### 3.2.1. Executive Functions

Three different executive functioning tests, each representing one domain of executive functioning, were used to investigate whether executive functioning can explain pain responses in the group of older individuals. Sex was added as a covariate given the found sex differences for pressure pain sensitivity. The average scores of the executive functioning tests are displayed in Table 2. Table 3 shows the results of the regression analyses.

Cognitive inhibition: The Stroop Interference scores, used to measure cognitive inhibition, could significantly explain variance in pain responses (multivariate outcome). Univariate tests showed that cognitive inhibition performance was primarily associated with pressure pain sensitivity measured via facial expression, with worse performance on the Stroop test predicting stronger facial responses during painful pressure stimulation.

Shifting: Similar to the Stroop test, performance on the Trail Making Test B, used to measure shifting, could significantly explain variance in pain responses (multivariate outcome). Univariate tests showed that shifting performance was also primarily associated with pressure pain sensitivity. Worse performance on the Trail Making Test predicted stronger facial responses during painful pressure stimulation.

Updating: The number of words produced during the Letter Fluency test, used to measure updating, could not significantly explain variance in pain responses (multivariate outcome).

#### 3.2.2. Gray Matter Volume

VBM was used to identify significant correlations between pain responses and regional gray matter volume. In the case of subjective and facial pressure pain sensitivity, sex was added as a covariate. At the whole brain level, no significant correlations were observed for any of the pain responses. Region of interest analyses revealed that CPM measured via facial expression correlates with regional gray matter volume in two adjacent clusters in the right vlPFC (FWE corrected, *p* < 0.05), as illustrated by the highlighted area in Figure 3 and reported in Table 4. Better endogenous pain inhibition was related to larger gray matter volume in this area.

#### 3.2.3. White Matter Structure

Probabilistic tractography was used to trace the white matter tracts connecting the PAG with the left and right dlPFC and vlPFC as well as the vmPFC. The mean FA from voxels within these tracts was extracted for each participant. Regression analyses including sex as a covariate revealed that pain sensitivity measured via facial expression could be explained by the mean FA within the pathway between the PAG and the vmPFC (*R*^2^ = 0.347, *p* < 0.01, standardized beta coefficient = 0.340). For CPM, both subjective and facial measures could be explained by white matter structure. Mean FA within the pathway between the PAG and the right dlPFC could significantly explain variance in CPM measured via verbal rating (*R*^2^ = 0.131, *p* = 0.016, and standardized beta coefficient = 0.362), while variance in CPM measured via facial expression could be explained by the mean FA between the PAG and left vlPFC (*R*^2^ = 0.095, *p* = 0.042, and standardized beta coefficient = 0.308). For both pathways, increased FA was associated with decreased endogenous pain inhibition. Figure 4 shows the relationship between FA and CPM in scatterplots.

### 3.3. Hierarchical Regression Analyses Explaining Variance in Pain Inhibition

The previous sections demonstrate that structural brain variations can significantly explain variance in CPM in older individuals. In two additional analyses, we wanted to further investigate these significant associations between structural brain variations and CPM inhibition.

In the first analysis (I), we tested whether the association prevailed even when controlling for a potential moderator variable, namely, executive functioning (executive functioning correlated with CPM between *r* = 0.33 and 0.25). Hierarchical regression analyses were, therefore, conducted. Table 5 shows that structural brain variations could still significantly predict CPM inhibition even when controlling for executive functioning. More precisely, these findings confirm that gray matter volume within the vlPFC and the structure of the white matter tracts connecting the PAG with the vlPFC and dlPFC can significantly explain variance in CPM beyond what can be explained by executive functioning.

In the second analysis (II), we wanted to investigate whether gray matter and white matter FA could provide complementary information for explaining variability of CPM. Entering gray and white matter in a forward regression analysis showed that both gray and white matter provide complementary information for variability of CPM (Δ*p* = 0.002).

## 4. Discussion

This study aimed to investigate whether pain processing in older individuals can be related to executive functioning performance and structural variations in the prefrontal cortex. Our sample of older individuals did not show significant pain inhibition as measured using CPM, supporting evidence from previous studies in older individuals that showed decrease pain inhibition in older compared to younger individuals. This pattern of pain responses could partly be explained by executive functioning performance as well as by gray and white matter structure within and towards the prefrontal cortex. The predictive power of structural brain variations was limited to CPM effects and prevailed when controlling for executive functioning. Thus, structural variations in the prefrontal cortex could explain CPM responses in older adults beyond what could be explained by executive functioning. We will discuss these findings in detail below.

### 4.1. Executive Functioning and Pain Responses in Older Individuals

A substantial amount of studies have reported a significant association between reduced executive functioning performance and increased pain responses. However, as we could show in a recently published systematic review, the overall strength of this association is only weak [48]. This might be due to the diversity regarding different types of experimental pain responses and executive functioning tests. The executive functioning domain cognitive inhibition was most frequently found to be associated with experimental pain responses but also the effect size of this association was only small. However, these findings are mostly based on studies investigating young adults. In contrast, findings of studies investigating older individuals suggest that the association between reduced executive functioning performance and increased pain responses might be more pronounced in older individuals [6,7,8,9].

The current study showed that reduced cognitive inhibition and shifting are associated with increased pain responses (mostly facial responses to pressure pain) in older individuals. Cognitive inhibition and shifting could explain approximately 20% of the variance in pain responses. Previous studies in older individuals mostly only investigated cognitive inhibition. These studies demonstrated that older individuals with lower performance on cognitive inhibition tasks show reduced endogenous pain inhibition [9], increased responses to pressure stimulation [7], and reduced tolerance of cold pain [6]. In the present study, we found that a reduction in cognitive inhibition in older individuals was especially associated with stronger facial responses to pressure stimulation. This finding is in line with previous findings in younger [22] and older individuals [7]. Here, individuals with a lower performance on cognitive inhibition also showed more facial responses during painful stimulation.

With regard to the executive functioning domain shifting, we found a similar relation with facial expressiveness, namely, older individuals with lower shifting performance showed more facial responses during pressure stimulation. Thus, worse shifting performance seems to be linked to a greater tendency for pain facilitation. This corroborates the findings of an association between cognitive inhibition and pain responses, given that the Trail Making Test B also involves some cognitive inhibitory functioning, i.e., the ability to stop responses to a number to be prepared to respond to a letter.

Several studies have pointed towards a specific association between cognitive inhibition and endogenous pain inhibition. However, as we could show in our review article, the overall effect size of an association between CPM and cognitive inhibition was only small [48]. The present study could also only demonstrate a small nonsignificant association between cognitive inhibition and CPM (*r*-values ≤ 0.21). It might be possible that cognitive inhibition is more strongly related to other forms of pain inhibition beyond CPM.

In sum, this study shows that poor executive functioning in older individuals can be linked to increased pain responses, but this association was not specific to one domain of executive functioning or one type of pain response.

### 4.2. Brain Structure and Pain Responses in Older Individuals

To date, a small number of studies have used functional imaging to investigate the neural base of pain responses in older individuals [49,50,51], but no study has used structural imaging to investigate whether structural brain variations are associated with pain responses in older individuals. Studies in young adults showed that individual differences in pain sensitivity [32,33], pain inhibition [52], and temporal summation [30] can be linked to individual differences in regional gray matter structure. In contrast, in the present study, pain sensitivity in older individuals was not significantly linked to differences in gray matter structure nor to differences in white matter structure. Only variance in CPM could specifically be explained by structural brain variations in gray and white matter.

CPM was measured both via pain ratings as well as via facial expressions in response to the test stimuli. The conditioning stimulus was only mildly painful and therefore, no facial expression occurred in response to the conditioning stimulus. Since the responses to the bilaterally presented test stimuli were averaged, we had no prediction about the side of the brain that would be associated with CPM. We found that a loss in CPM measured via facial expression (meaning that the facial response to pressure stimulation did not decrease due to tonic heat counter stimulation) was related to reduced gray matter volume in the right vlPFC as well as to increased FA of the white matter tracts connecting the PAG with the left vlPFC. Moreover, CPM measured via verbal rating was related to increased FA of the white matter tracts connecting the PAG with the right dlPFC.

The importance of the prefrontal cortex in pain inhibition has been demonstrated by previous neuroimaging studies in younger healthy individuals. Several of these studies showed that the magnitude of the CPM effect is related to the level of activity in prefrontal areas during a CPM paradigm [29,34,53,54]. Furthermore, the thickness of the orbitofrontal cortex has been found to correlate with CPM [52]. In these young adults, a reduced response to a test stimulus presented on the right leg in the presence of a conditioning stimulus presented on the contralateral arm was correlated with a thicker right lateral orbitofrontal cortex. This is partly in line with our finding that CPM is associated with gray matter volume in the right vlPFC, because this area is close to the right lateral orbitofrontal cortex. We found that in older individuals, reduced CPM is associated with reduced gray matter volume in the right vlPFC. This is in line with the notion that age-related reduction in prefrontal volume might lead to reduced pain inhibition and thus, to higher prevalence of pain.

With regard to white matter, there is only one previous study that investigated the association between white matter structure and endogenous pain modulation [40]. They showed that stronger placebo analgesia responses are associated with increased FA in the white matter tracts between the PAG and the dlPFC. Although CPM is not directly related to the level of placebo analgesia in an individual [55], we expected that similar white matter tracts might play a role in CPM. We indeed found significant associations, however, contrary to expectations, we found that increased FA of the white matter tracts between the PAG and the prefrontal cortex was not associated with better pain inhibition, but with worse pain inhibition. When interpreting the direction of FA, there is the difficulty that FA is influenced by several factors, including myelination, axon diameter, and the space between axons [56]. Nevertheless, in general, increased FA is thought to reflect better white matter structure. We can only speculate why increased FA was associated with poorer CPM. One possibility might be that loss in gray matter in older individuals might be compensated by an increase in white matter connections. We also might have missed relevant white matter tracts by focusing solely on prefrontal connections. A recent study investigated the neural correlates of CPM beyond the prefrontal cortex using resting functional connectivity, which measures the temporal correlation between brain activity patterns of different regions [57]. Better pain inhibition was found to be associated with higher functional connectivity between the PAG and areas in the anterior cingulate cortex, insula, pons, and rostroventromedial medulla. Future studies might investigate whether structural variations in these white matter tracts beyond the prefrontal cortex might also explain variance in endogenous pain inhibition in older individuals.

In summary, we found that out of all pain measures, only CPM was related to structural brain variations in gray and white matter. This association prevailed even when controlling for executive functioning. Together, structural variations in the prefrontal cortex and executive functioning performance could explain 37% and 22% of the variance in CPM measured by facial expression and verbal rating, respectively.

### 4.3. Limitations

This study is a first explorative attempt to understand whether structural brain variations in the prefrontal cortex can explain pain responses in older individuals. The study should be replicated in another age group to examine whether our findings are unique to older individuals. In line with this, we cannot be certain that the deficits in pain inhibition measured with CPM protocol were indeed due to age, or might be due to the experimental design, given that we did not include a young control group. Furthermore, we used pain protocols that should be suitable for various groups of older individuals with various degrees of cognitive capacities. Therefore, we decided to use a cognitively less challenging rating scale and to use fixed stimulus intensities instead of intensities tailored to the individual pain threshold because assessment of pain threshold is not possible in all groups of older individuals (e.g., individuals with aphasia or dementia). Using less individualized pain intensities might have increased error variance in our sample. Finally, the small sample size limits the generalizability of the findings of this study.

## 5. Conclusions

The present study showed that pain responses observed in older individuals are associated with executive functioning performance and variations in gray and white matter within and towards the prefrontal cortex. Specifically, variance in CPM was associated with these structural brain variations. Speculatively, this association could be one of the underlying mechanisms explaining age-related changes in pain processing and therefore, increased clinical pain prevalence.

## Figures and Tables

**Figure 1 brainsci-10-00477-f001:**
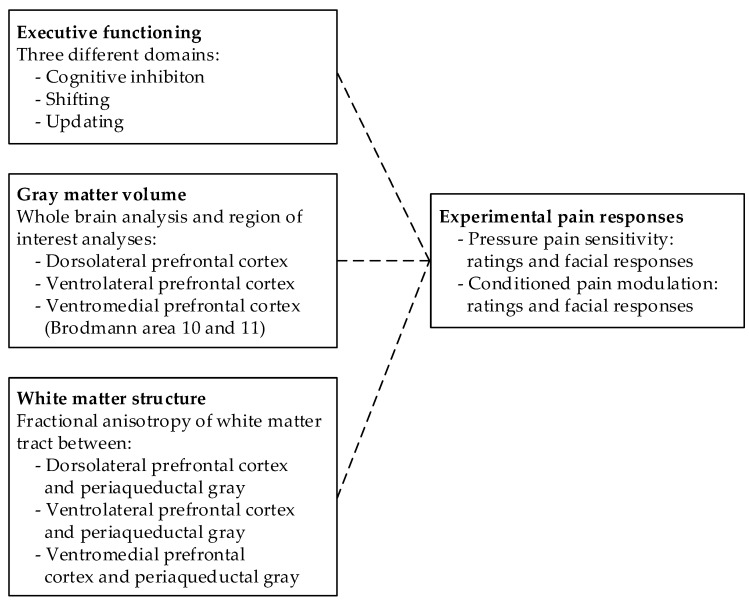
An overview of all variables used in this study.

**Figure 2 brainsci-10-00477-f002:**
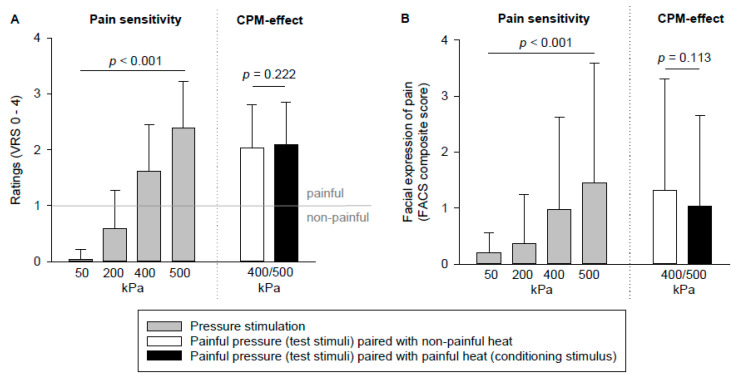
Average pain ratings (**A**) and the Facial Action Coding System (FACS) composite scores (**B**) of the different experimental pain stimuli. Subjective and facial responses to these stimuli significantly increased across intensities. No significant conditioned pain modulation (CPM) effect was found, neither when examining pain ratings nor when examining facial responses. CPM, conditioned pain modulation; kPa, kilopascal; FACS, facial action coding system; VRS, verbal rating scale (0 = no pain, 1 = mild pain, 2 = moderate pain, 3 = strong pain, and 4 = very strong pain).

**Figure 3 brainsci-10-00477-f003:**
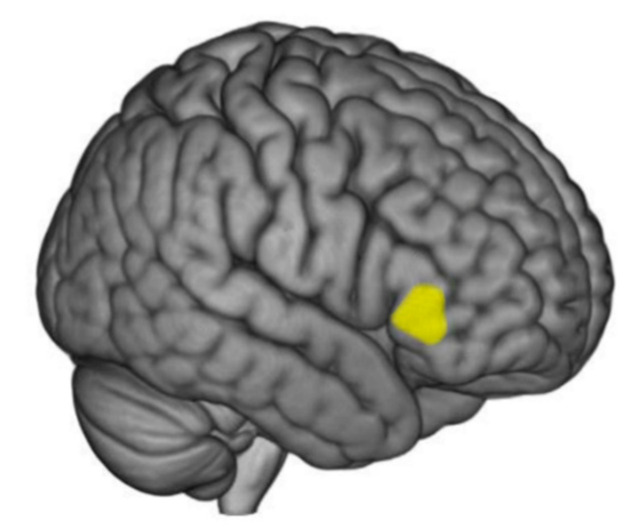
Better endogenous pain inhibition (CPM) was related to larger gray matter volume in the highlighted area in the right ventrolateral prefrontal cortex. See Table 4 for further details.

**Figure 4 brainsci-10-00477-f004:**
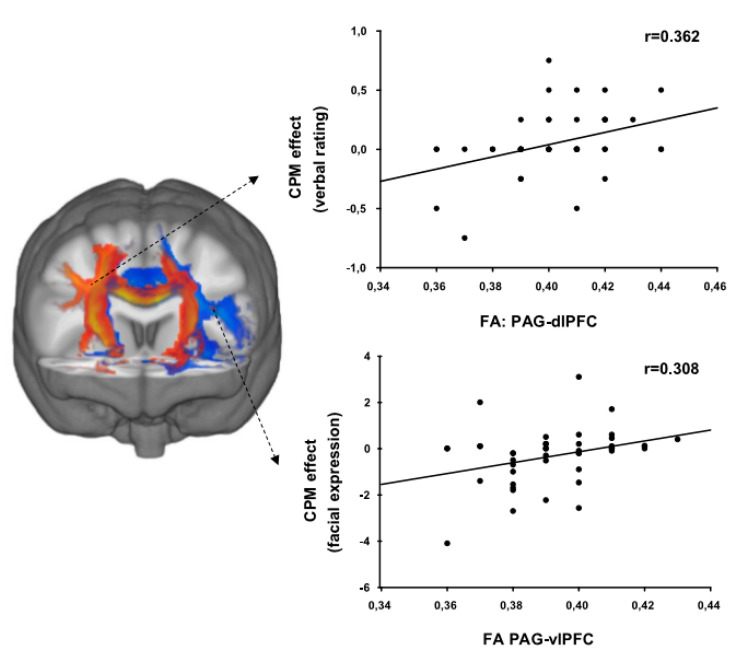
The level of pain inhibition (CPM) is significantly associated with the structure of the white matter tracts connecting the PAG with the right dorsolateral prefrontal cortex (orange) and the left ventrolateral prefrontal cortex (blue). An example of the pathways is shown for one participant overlaid on the MNI152 standard brain (coronal view; y = 15). Mean FA within the pathway connecting the PAG with the right dorsolateral prefrontal cortex correlates with pain inhibition measured via verbal rating. Mean FA within the pathway connecting the PAG with the left ventrolateral prefrontal cortex correlates with pain inhibition measured via facial expression. For both pathways, increased FA was associated with decreased pain inhibition. CPM, conditioned pain modulation; FA, fractional anisotropy.

**Table 1 brainsci-10-00477-t001:** Facial action units (AUs) selected for further analysis.

Action Unit	Description	Percentage *
AU 1/2	Brow raiser	18.1
AU 4	Brow lower	14.1
AU 6/7	Orbit tightening	39.9
AU 9/10	Levator contraction	20.3
AU 25/26/27	Mouth opening	35.1

* Cumulative percentage of occurrence during all 500 kPa pressure stimuli.

**Table 2 brainsci-10-00477-t002:** Demographic characteristics and executive functioning performance.

	Male (*N* = 25)	Female (*N* = 21)	Statistics
*Demographic characteristics*			
Age	67.8 ± 5.6 (mean ± SD)	67.5 ± 6.7 (mean ± SD)	*t* (44) = 0.2, *p* = 0.88
Education	*N* (%)	*N* (%)	
High school	4 (16)	7 (33)	X^2^(3) = 1.9, *p* = 0.60
Secondary vocational education	4 (16)	2 (10)	
Higher professional education	12 (48)	9 (43)	
University education	4 (16)	3 (14)	
*Executive functioning performance*	mean ± SD	mean ± SD	
Stroop Interference score	30.3 ± 10.7	29.7 ± 10.2	*t* (44) = 0.2, *p* = 0.84
Trail Making Test Part B (time in seconds)	73.6 ± 20.0	84.9 ± 55.9	*t* (43) = 0.4, *p* = 0.35
Letter Fluency test (total number of words)	13.6 ± 4.6	16.3 ± 4.8	*t* (44) = −2.0, *p* = 0.06
*Self-Report Rating*	median [IQR]	median [IQR]	
50 kPa rating	0 [0–0]	0 [0–0]	F(1,44) = 6.5, *p* = 0.015
200 kPa rating	0 [0–1]	1 [0–1.5]	
400 kPa rating	1 [1–2]	2 [1–2.75]	
500 kPa rating	2 [2–2.75]	2.5 [2–3.5]	
*Facial responses*	mean ± SD	mean ± SD	
50 kPa facial response	0.20 ± 0.41	0.19 ± 0.33	F(1,44) = 14.33; *p* < 0.001
200 kPa facial response	0.15 ± 0.31	0.61 ± 1.23	
400 kPa facial response	0.26 ± 0.60	1.83 ± 2.07	
500 kPa facial response	0.59 ± 1.17	2.47 ± 2.57	

kPa, kilopascal; IQR, interquartile range; SD, standard deviation.

**Table 3 brainsci-10-00477-t003:** Multiple regression analyses predicting pain responses by executive functioning performance.

		*R* ^2^	*p*	Standardized Beta Coefficients
*Stroop Interference score*				
Multivariate outcome		0.263	0.027 *	
Univariate tests	Sex (covariate)		0.088	−0.295
	Pressure pain sensitivity: verbal rating		0.141	0.228
	Pressure pain sensitivity: facial expression		0.028 *	0.374
	Pain inhibition: verbal rating		0.181	0.193
	Pain inhibition: facial expression		0.082	−0.250
*Trail Making Test B*				
Multivariate outcome		0.225	0.067	
Univariate tests	Sex (covariate)		0.827	−0.038
	Pressure pain sensitivity: verbal rating		0.225	−0.195
	Pressure pain sensitivity: facial expression		0.003 *	0.530
	Pain inhibition: verbal rating		0.740	0.049
	Pain inhibition: facial expression		0.643	−0.068
*Letter Fluency test*				
Multivariate outcome		0.182	0.139	

Multivariate regression analyses were conducted to examine whether variance in overall pain responses could be explained by executive functioning performance. Univariate tests were used to examine which specific pain response could be predicted by the executive functioning tests. * *p* < 0.05.

**Table 4 brainsci-10-00477-t004:** Clusters of voxels that showed a significant correlation with pain inhibition (conditioned pain modulation, CPM) measured via facial expression in a region of interest VBM analysis (FWE-corrected, *p* < 0.05).

Anatomical Region	Side	Brodmann Area (BA)	MNI Peak Coordinate (x, y, z)	Peak *T* Value	Cluster Size (Number of Voxels)
Ventrolateral Prefrontal Cortex	Right	BA45	38, 26, 2	4.24	30
	Right	BA47	36, 30, −2	4.21	4

Better inhibition was related to larger regional gray matter volume in two adjacent clusters in the right ventrolateral prefrontal cortex. Cluster size in voxels at an extent threshold of *p* < 0.001 uncorrected. FWE, family-wise error; MNI, Montreal Neurological Institute; VBM, voxel-based morphometry.

**Table 5 brainsci-10-00477-t005:** Hierarchical regression analyses of predictors of pain inhibition (CPM).

	Model	*R* ^2^	*p*	Δ*R*^2^	Δ*p*
Pain Inhibition Facial Expression	1	0.065	0.444		
	2a	0.366	0.004 *	0.300	0.001 *
Pain Inhibition Verbal Rating	1	0.109	0.207		
	2b	0.223	0.044 *	0.113	0.024 *

Model 1: executive functioning (Stroop Interference score, Trail Making Test Part B, and Letter Fluency test) included. Model 2a: executive functioning, volume of Brodmann area 45 and mean FA between the PAG and left vlPFC included. Model 2b: executive functioning and mean FA between the PAG and right dlPFC included * *p* < 0.05.

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
