# Peer review of "Pain Processing in Older Adults and Its Association with Prefrontal Characteristics"

_brainsci, 2020, doi:10.3390/brainsci10080477_

Round 1

Reviewer 1 Report

This study aimed to investigate whether pain responses in older individuals are associated with prefrontal characteristics, namely a) executive functioning performance and b) structural brain variations in the prefrontal cortex. To do this, the authors assessed pain responses (e.g., pressure pain sensitivity and endogenous pain inhibition) in 46 healthy older individuals and investigated whether these pain responses were associated with a) different domains of executive functioning and b) to variations in gray and white matter structure of the prefrontal cortex. Overall pain responses were significantly associated with the executive functioning domains cognitive inhibition and shifting. Specifically, endogenous pain inhibition showed a significant association with gray matter volume in the prefrontal cortex and with variations in white matter structure of tracts connecting the prefrontal cortex with the periaqueductal gray. Speculatively, this association might explain age-related changes in pain processing and therefore increased clinical pain prevalence.

Overall, this is a well-described study. The manuscript is also well written. The aims and hypothesis are clearly stated and the results are fairly conclusive. I approve this manuscript.

Author Response

1) Overall, this is a well-described study. The manuscript is also well written. The aims and hypothesis are clearly stated and the results are fairly conclusive. I approve this manuscript.

Answer: We thank the reviewer for the positive feedback.

Reviewer 2 Report

Summary: The article titled Pain processing in older adults and its association with prefrontal characteristics, is an investigation of pressure pain sensitivity and conditioned pain modulation among healthy older adults, and their relationship to executive functioning performance and structural brain variations in whole brain and regions of interest (e.g., prefrontal). Regression analyses were conducted to assess the relationships. Pain was reported using both verbal ratings and facial patterns.

Concerns/Comments: I commend the authors for taking on this topic. This is a very timely and relevant topic that needs to be addressed in current pain research. I think this article will make an important contribution to the literature. However, I have some major concerns I outline below. I hope that my comments provide constructive feedback that can be used to refine this manuscript.

Major

1) Pain sensitivity was rated using an ordinal variable, with 5 possible indicators (no pain, mild pain, moderate pain, strong pain, and very strong pain. On page 8, lines 292-294, it states that the average responses to the 400 and 500 kPa stimuli were both rated as painful (verbal rating scale ≥ 1), yet one could argue that “mild pain” is very different from “very strong pain”. Can you give an argument for why this was done? Looking at Figure 2a, I get the impression that none of the trials were very painful, that most people were reporting 400 and 500 kPa as mild to moderate pain. If this is true, then one could argue the point that due to low variability in pain ratings to the different stimuli, these were averaged. Yet, I don’t see this type of argument or any descriptive statistics that would lead me to that conclusion.

2) Verbal pain ratings are reported using the average? I am assuming this is the mean? That is, the ordinal pain sensitivity variable is being treated as a continuous variable. Providing median (IQR) for each pain rating would be helpful.

3) Full information from the ANOVA and t test analysis would be helpful. A table that included this information as well as the median (IQR) for pain ratings would provide the reader with a better sense of pain responses in this group.

4) In order to better understand the relationships being presented, coefficients are needed in the Tables, including covariates.

5) Executive functioning outcomes are presented by sex (males vs. females). Can the authors explain why this variable (sex) was chosen? Also, in Table 2, is the difference on the Trail Making Test Part B significant? The text mentions that the proportion of men and women is balanced, but nothing else is said about potential sex differences in cognition, pain, or brain structure. It might be informative to present descriptive statistics for the these tasks based on pain reporting (pain vs. no pain). In particular, one might expect sex stratified analyses given that the Results begin with descriptives broken out by sex. IF there are no sex differences in cognitive performance, then saying so would be helpful. Then the reader would understand explicitly why this variable was not entered as a covariate in the models, given recognized differences in pain, cognition, and brain structure across sex.

6) Were any other covariates included, such as age, education, income? This is particularly relevant to a study investigating pain and executive functioning. If these demographics were assessed, descriptive statistics needs to be included and a justification for why they were not entered in the models. The lack of attention to these variables in this paper gives me sufficient concern regarding interpretation of the findings (see Poleshuck and Green, 2008, Pain, “Socioeconomic Disadvantage and Pain”, among others).

7)Discussion section: The Discussion section is a well written synthesis of the literature. However, it is worth mentioning in the limitations section the lack of other potential covariates in the models and the small sample size limiting generalizability.

Author Response

1) Pain sensitivity was rated using an ordinal variable, with 5 possible indicators (no pain, mild pain, moderate pain, strong pain, and very strong pain. On page 8, lines 292-294, it states that the average responses to the 400 and 500 kPa stimuli were both rated as painful (verbal rating scale ≥ 1), yet one could argue that “mild pain” is very different from “very strong pain”. Can you give an argument for why this was done? Looking at Figure 2a, I get the impression that none of the trials were very painful, that most people were reporting 400 and 500 kPa as mild to moderate pain. If this is true, then one could argue the point that due to low variability in pain ratings to the different stimuli, these were averaged. Yet, I don’t see this type of argument or any descriptive statistics that would lead me to that conclusion.
Answer: We thank the reviewer for pointing this out. It is correct that none of the trials were rated as very painful. Only 400 and 500 kPa elicited stable painful ratings that varied between mild to moderate; as the reviewer has pointed out. This low variability in pain ratings was indeed the reason that we averaged the responses to 400 and 500 kPa. We now describe this more carefully in the manuscript (line 301-303).

 2) Verbal pain ratings are reported using the average? I am assuming this is the mean? That is, the ordinal pain sensitivity variable is being treated as a continuous variable. Providing median (IQR) for each pain rating would be helpful.
Answer: We agree with the reviewer that providing the median and IQR is important. We now provide these descriptive statistics in Table 2.   

 3) Full information from the ANOVA and t test analysis would be helpful. A table that included this information as well as the median (IQR) for pain ratings would provide the reader with a better sense of pain responses in this group.
Answer: We thank the reviewer for this suggestion. Table 2 now includes all test statistics, as well as the median (IQR) of the pain ratings.

 4) In order to better understand the relationships being presented, coefficients are needed in the Tables, including covariates.
Answer: We thank the reviewer for this suggestion, beta coefficients are now provided for all regression analyses (Table 3, line 369-374).

 5) Executive functioning outcomes are presented by sex (males vs. females). Can the authors explain why this variable (sex) was chosen? Also, in Table 2, is the difference on the Trail Making Test Part B significant? The text mentions that the proportion of men and women is balanced, but nothing else is said about potential sex differences in cognition, pain, or brain structure. It might be informative to present descriptive statistics for the these tasks based on pain reporting (pain vs. no pain). In particular, one might expect sex stratified analyses given that the Results begin with descriptives broken out by sex. IF there are no sex differences in cognitive performance, then saying so would be helpful. Then the reader would understand explicitly why this variable was not entered as a covariate in the models, given recognized differences in pain, cognition, and brain structure across sex.
Answer: We apologize for the lack of clarity on our side. As a first step, we examined whether pain responses were affected by potential confounders, namely age and sex, which are known to have an influence on pain. We now also included the level of education, following remark 5 of reviewer 1 (see comment below). Whereas education and age had no impact on the pain responses in our sample,  sex significantly affected pain responses. We therefore present demographics, executive functioning performance and pain responses by sex (Table 2). Executive functioning did not differ between males and females. This is now stated more clearly in Table 2.

 6) Were any other covariates included, such as age, education, income? This is particularly relevant to a study investigating pain and executive functioning. If these demographics were assessed, descriptive statistics needs to be included and a justification for why they were not entered in the models. The lack of attention to these variables in this paper gives me sufficient concern regarding interpretation of the findings (see Poleshuck and Green, 2008, Pain, “Socioeconomic Disadvantage and Pain”, among others).
Answer: We thank the reviewer for this important comment. Following the suggestion, we now checked whether pain responses in the present sample were indeed affected by education. As stated above, education did not significantly impact the pain responses in our study (line 292). The difference to the cited paper by Poleshuck and Green 2008 and our study is, that we focused on experimental pain responses and not on chronic pain conditions.

 7)Discussion section: The Discussion section is a well written synthesis of the literature. However, it is worth mentioning in the limitations section the lack of other potential covariates in the models and the small sample size limiting generalizability.
Answer: We think we now explain more carefully how we controlled for potential covariates in the manuscript. We added the sample size argument to the limitation section (line 525-526).

Reviewer 3 Report

The authors report pain response in older healthy individuals and try to link the pain processing into executive functioning performance and structural variations in the prefrontal cortex. The topic is interesting and the methods are good. Further, the author also provides predictors used for hierarchical regression analyses of pain inhibition.

However, as the authors mentioned, they did not include a young controls, so it is not convincible to make some of the conclusions, for example “older adults show deficits in pain inhibition and a trend towards pain facilitation”. Also, these are all healthy older individuals, it is hard to make a conclusion about the association between pain inhibition and structure variations in the prefrontal cortex.

It is very nice that the study includes both male and females. It seems no difference in executive functioning performance in both sex, is there any difference in the gray matter volume or white matter structure between male and females.

It would be helpful that the authors provide more details for the pain measurement. For example, did you do repeat measure for individuals? If so, how many repeats? Was the measure carried out in the morning or in the afternoon?

Author Response

1) However, as the authors mentioned, they did not include a young controls, so it is not convincible to make some of the conclusions, for example “older adults show deficits in pain inhibition and a trend towards pain facilitation”. Also, these are all healthy older individuals, it is hard to make a conclusion about the association between pain inhibition and structure variations in the prefrontal cortex.
Answer: We thank the reviewer for pointing this out. We now try to be more cautious in our conclusions. For example, we now toned down that sentence (line 411-414), as well as the concluding sentence of the abstract (line 32-33) and the conclusion (line 528-531).

2) It is very nice that the study includes both male and females. It seems no difference in executive functioning performance in both sex, is there any difference in the gray matter volume or white matter structure between male and females.
Answer: We indeed found no difference in executive functioning performance between males and females, which we now made more clear in Table 2. We now also included sex as a covariate in all analyses including facial and subjective responses to pressure pain (given that sex differences were found for these dependent variables). This also includes the brain analyses.

We also checked whether the regions of interest that showed associations with the pain responses  differed between males and females. As can be seen in the Table below, we did not find differences in gray matter volume or white matter structure between male and females in our regions of interest (t-tests).

Male

(mean ± SD)

Female

(mean ± SD)

p-value

GM volume brodmann area 45

1.30 ± 0.12

1.29 ± 0.16

0.86

Mean FA right vlPFC

0.39 ± 0.02

0.38 ± 0.02

0.17

Mean FA right vlPFC

0.40 ± 0.02

0.39 ± 0.02

0.20

Mean FA right vlPFC

0.41 ± 0.02

0.40 ± 0.02

0.16

Mean FA right vlPFC

0.40 ± 0.02

0.40 ± 0.02

0.32

Mean FA right vlPFC

0.35 ± 0.02

0.35 ± 0.02

0.43

3) It would be helpful that the authors provide more details for the pain measurement. For example, did you do repeat measure for individuals? If so, how many repeats? Was the measure carried out in the morning or in the afternoon?
Answer: We apologize for not reporting this information. Each measure was repeated twice, once to the right shoulder and once to the left shoulder, which is described at line 101-102. We now also added to the manuscript what time of the day the pain measurements were conducted (line 87).

Round 2

Reviewer 2 Report

Thank you for responding to my comments.

Reviewer 3 Report

The author has answered my concerns. The MS has been improved.